# Fast Automatic Fuzzy C-Means Knitting Pattern Color-Separation Algorithm Based on Superpixels

**DOI:** 10.3390/s24010281

**Published:** 2024-01-03

**Authors:** Xin Ru, Ran Chen, Laihu Peng, Weimin Shi

**Affiliations:** College of Mechanical Engineering, Zhejiang Sci-Tech University, Hangzhou 310018, China

**Keywords:** knitting CAD, color-separation algorithm, blind super-resolution network, superpixel algorithm, density peak clustering (DPC), fast fuzzy c-means (FCM)

## Abstract

Patterns entered into knitting CAD have thousands or tens of thousands of different colors, which need to be merged by color-separation algorithms. However, for degraded patterns, the current color-separation algorithms cannot achieve the desired results, and the clustering quantity parameter needs to be managed manually. In this paper, we propose a fast and automatic FCM color-separation algorithm based on superpixels, which first uses the Real-ESRGAN blind super-resolution network to clarify the degraded patterns and obtain high-resolution images with clear boundaries. Then, it uses the improved MMGR-WT superpixel algorithm to pre-separate the high-resolution images and obtain superpixel images with smooth and accurate edges. Subsequently, the number of superpixel clusters is automatically calculated by the improved density peak clustering (DPC) algorithm. Finally, the superpixels are clustered using fast fuzzy c-means (FCM) based on a color histogram. The experimental results show that not only is the algorithm able to automatically determine the number of colors in the pattern and achieve the accurate color separation of degraded patterns, but it also has lower running time. The color-separation results for 30 degraded patterns show that the segmentation accuracy of the color-separation algorithm proposed in this paper reaches 95.78%.

## 1. Introduction

In the traditional textile industry, the pattern design of knitted products requires manual tracing, which is time-consuming and cannot easily achieve the desired results. With the development of the industry, we now produce and simulate knitted product pattern designs through knitting CAD (computer-aided design) [1] to show the characteristics of the knitted fabric’s shape, color, structure, etc., which improves the design of the product and creates favorable conditions for the mass production of textile products and adapting them to meet market demand.

Color separation [2], also known as color segmentation, is an important image processing function of knitting CAD, in which the original colorful pattern is represented by a small number of colors without changing the image effect. The patterns [3] that we obtain through technical means, such as internet downloads, scanners, digital cameras, etc., are true color images with thousands or tens of thousands of different colors. However, the variety of yarn colors for pattern knitting is limited, so we must use the color-separation function of knitting CAD to merge similar colors from the images to generate patterns. Although knitting CAD saves much time compared with traditional pattern design methods, the current color-separation algorithm cannot achieve the ideal processing effect for degraded patterns. The resulting products have defects such as variegated spots, inaccurate edges, and color errors, and the parameter of the number of clusters needs to be set manually [4], which does not enable complete automatic color separation.

Patterns exhibit different degrees of quality degradation in the processes of image shooting, network transmission, image compression, and image editing [5], so most of the patterns we see are not particularly clear. Figure 1a,b shows degraded patterns and the number of colors in RGB space they possess; it can be seen that the color boundaries of the patterns are blurred, and the number of colors in the patterns far exceeds the range of the number that can be recognized by the human eye. There are many research methods for improving image quality, including the super-resolution reconstruction network [6], which mainly recovers low-resolution images by learning the mapping relationship between low- and high-resolution images [7]. However, the degradation of real images usually occurs due to a complex combination of different degradation processes, and these methods [8,9,10,11,12] only assume an ideal bicubic downsampling degradation kernel, which makes it difficult to simulate complex degradation situations. Wang, X. et al. proposed a Real-ESRGAN blind super-resolution network [13], which expands the classical “first-order” degradation model to a “high-order” degradation model for real-world degradations, i.e., the degradations are modeled with several repeated degradation processes, with each process represented by the classical degradation model. This method better simulates some unknown and complex degradation processes in real images, thus recovering their degradation patterns. However, as image resolution increases, the number of pixels and colors also increases, which greatly increases the computation complexity of the color-separation algorithm.

According to the requirements of knitting product manufacturing processes, where possible, color separation needs to be performed without affecting the perception of the human eye, i.e., while maintaining maximum similarity between the color-separated image and the original image, and the edges need to be as smooth and accurate as possible, with no variegated spots in one color region. Currently, pixel-based clustering algorithms are commonly used for image color separation, such as K-means [14], hierarchical clustering [15], spectral clustering [16], fuzzy c-mean (FCM) clustering [17], self-organizing mapping neural networks (SOM) [18], and deviation-sparse fuzzy c-means with a neighbor information constraint (DSFCM_N) [19]. However, these algorithms only perform unsupervised classification of a large number of pixels based on color similarity information, ignoring the local spatial information of pixels, and are prone to color-separation errors. In particular, the color of the edge part is formed by mixing two colors at the junction; the color of the edge is completely different from the color of the central pattern, and the color-separation result will show variegated spots. Moreover, all these algorithms require predefined parameters for the number of clusters, which considerably affects their productivity. The number of pixels in an image is elevated after clarification, and if the above algorithms are used to process high-resolution images, the computational overhead is high because repeated computation and iterative optimization of the same pixels are required [20]. Rodriguez, A. et al. proposed density peak clustering (DPC) [21], which automatically calculates the number of clusters for the data points, but the algorithm produces a large similarity matrix when processing images directly, leading to memory overflow. Zhang, J. et al. proposed a clustering algorithm that combines SOM and DPC [22], where the SOM algorithm identifies the main image color clusters and then the best number of colors, and the clusters are merged by the DPC algorithm; this algorithm shortens the computation time, but the same problem of variegated spots will occur. Qian, M. [23] et al. proposed a color-segmentation algorithm combining SOM and efficient dense subspace clustering (EDSC) and eliminated the mis-segmentation of edge colors through post-processing techniques such as gray-scale transformation, binarization, and open operation; however, the accuracy of this method is still not high, and the process is complicated. Kumah, C. et al. used an unsupervised mean shift algorithm for the color segmentation of printed fabrics [24]. The experimental results show that the algorithm is able to accomplish the color segmentation of fabric images under different texture and illumination conditions. The algorithm has strong robustness but has some shortcomings in industrial applications because fabrics need to be cleaned and ironed before color segmentation.

Superpixel segmentation [25] as a color-separation preprocessing method can effectively reduce the complexity of image information [26]. Superpixel segmentation uses a small number of pixel blocks instead of a large number of pixel points to express image features, by combining the color information and spatial information of the pixels [27], and enhances the accuracy of the color-separation algorithm. Achanta, R. et al. proposed a simple linear iterative clustering (SLIC) algorithm [28], which generates compact, approximately uniform hexagonal superpixel regions with few parameter settings. However, the edges of superpixel images generated by this algorithm are not smooth, and there is a problem of over-segmentation. Hu, Z. et al. proposed a watershed transform (WT) algorithm [29], which is an image-segmentation algorithm based on the idea of mathematical morphology and of geomorphology and region growth. The algorithm produces irregular superpixels more effectively than SLIC, but it is prone to noise interference and exhibits considerable over-segmentation. Lei, T. et al. proposed a superpixel-based fast fuzzy c-means (SFFCM) algorithm [30], in which watershed transform based on multiscale morphological gradient reconstruction (MMGR-WT) is used as a pre-color-separation process, which overcomes the problem of over-separation and obtains a superpixel image with smooth edges without additional parameter settings. This method is based on the contour information of the gradient image for superpixel segmentation, and there is no edge color mis-segmentation, which solves the problem of variegated spots. However, the contour information of the image is calculated based on the difference between the erosion and dilation operations, and detailed areas in the image are often removed as noise, which can lead to the loss of some information in the image [31].

In order to perform automatic and accurate color separation of degraded patterns, this paper proposes a fast, automatic FCM color-separation algorithm based on superpixels. For the first time, we applied the blind super-resolution network and superpixel algorithm to color separation, which can achieve the automatic color separation of degraded patterns with high segmentation accuracy at a very low computational cost. The contribution of this paper is three-fold: first, we use the Real-ESRGAN blind super-resolution network as an image clarification step, which can obtain higher-resolution images and enhance the accuracy of the color-separation algorithm; second, we propose an improved MMGR-WT superpixel algorithm for generating superpixel images with smooth and accurate edges. This algorithm can effectively simplify the image information of high-resolution patterns, reduce the computation time of subsequent clustering, solve the problem of variegated spots, and does not lose information; finally, in order to achieve automatic color separation of images, we propose an improved DPC algorithm for calculating the number of superpixel clusters, and the number of clusters is consistent with the perception of the human eye.

## 2. Materials and Methods

The flow of the color-separation algorithm proposed in this paper for degraded patterns is shown in Figure 2, which includes four main steps: clarification processing, superpixel segmentation, superpixel clustering, and image deflation. First, the degraded patterns are clarified using the Real-ESRGAN blind super-resolution network [13] to obtain high-resolution color images with clear boundaries; the clarification results and the number of colors in RGB space are shown in Figure 1c,d. Then, the improved MMGR-WT superpixel algorithm is used to perform superpixel segmentation on the high-resolution images, and superpixel images with smooth and accurate edges are obtained. The improved DPC algorithm is then applied to the superpixels to automatically determine the number of clusters of superpixels without affecting the perception of the human eye, and the superpixel clustering is completed via fast FCM based on a color histogram. Since the Real-ESRGAN blind super-resolution network changes the size of the original image, the nearest-neighbor interpolation algorithm [32] needs to be used in the final step to resize the color-separated image, and this method does not change the number of colors in the image. In this paper, an improved MMGR-WT superpixel algorithm, an improved DPC algorithm, and fast FCM clustering based on a color histogram are introduced.

### 2.1. Improved MMGR-WT Superpixel Algorithm

The detail information in a pattern is very rich, and if the MMGR-WT superpixel algorithm is used for the superpixel segmentation of a high-resolution pattern, detail information loss will occur. In this paper, bilateral filtering [33] is applied to the MMGR-WT superpixel algorithm, and the algorithm framework is shown in Figure 3. First, the image is processed through bilateral filtering to enhance the edge information of the image; then, a gradient image is obtained by the Sobel operator [34], and a multiscale morphological gradient reconstruction (MMGR) operation is defined to obtain a contour-accurate superpixel image after WT. Finally, a color histogram of the superpixel image is created.

Bilateral filtering combines the spatial proximity information and color similarity information of pixels in a neighborhood and achieves filtering by applying a convolution operation to the pixels in the neighborhood of the image and the weight coefficients of the filter. While filtering out the noise and smoothing the image, the edge information of the image is retained so that the gradient information is more accurate and continuous. Let *f*(*x*,*y*) denote the pixel value of an image *f* at (*x*,*y*); then, the expression for bilateral filtering is:(1)$$f(x,y)=\frac{\sum_{\left(i,j)\in R(m,n\right)}f(i,j)w(x,y,i,j)}{\sum_{\left(i,j)\in R(m,n\right)}w(x,y,i,j)},$$
where *R*(*m*,*n*) is the neighborhood with a size of (2*m* + 1) × (2*n* + 1) around the output pixel point, and *w*(*x*,*y*,*i*,*j*) is the weight coefficient of each pixel in the neighborhood. The weight coefficients are composed of the product of the distance template coefficients *w_d_*(*x*,*y*,*i*,*j*) and the value domain template coefficients *w_r_*(*x*,*y*,*i*,*j*).
(2)$$w(x,y,i,j)=w_{d}(x,y,i,j)\times w_{r}(x,y,i,j),$$
(3)$$w_{d}(x,y,i,j)=e^{\left(-\frac{{\left(x-i\right)}^{2}+{\left(y-j\right)}^{2}}{2\sigma_{d}^{2}}\right)},$$
(4)$$w_{r}(x,y,i,j)=e^{\left(-\frac{{\left\|f(x,y)-f(i,j)\right\|}^{2}}{2\sigma_{r}^{2}}\right)}.$$

In the above equations, *σ* is the standard deviation of the Gaussian function.

The direct use of the WT algorithm for gradient images causes severe over-segmentation, and morphological gradient reconstruction (MGR) [35] is effective in overcoming this problem. However, the structuring element (SE) of MGR has a single size and cannot adaptively meet the needs of different images. MMGR reconstructs the gradient image by fusing the multi-size SE values and eliminates the dependence of segmentation results on SE values. By calculating the pointwise maximum of multiple reconstructed images, most of the useless local minima are removed while important edge details are preserved. MMGR is defined using *R^MC^* as follows:(5)$$R_{f}^{MC}(g,r_{1},r_{2})=\vee \{R_{f}^{C}{\left(g\right)}_{B_{r_{1}}},R_{f}^{C}{\left(g\right)}_{B_{r_{1}+1}},\cdots ,R_{f}^{C}{\left(g\right)}_{B_{r_{2}}}\},$$
where *B* is the structure element SE with radius *r*; *r*_1_ and *r*_2_ represent the minimum and maximum values of *r*, i.e., *r*_1_ ≤ *r* ≤ *r*_2_, *r*_1_, *r*_2_ ∈ N+; *f* is the input image; *g* is the labeled image; ∨ represents the pointwise maximum; and $R_{f}^{C}{\left(g\right)}_{B}$ denotes the morphological closed reconstruction, i.e., the addition of erosion reconstruction on top of the closed operation, defined as:(6)$$R_{f}^{c}{\left(g\right)}_{B}=R^{\varepsilon }(f•B),$$
where • is the morphological closed operation, *ε* represents the erosion operation, and the corresponding *R^ε^* represents the erosion reconstruction.

By combining bilateral filtering and MMGR-WT, a superpixel image with smooth and accurate edges can be obtained in a shorter time. Then, the pixel values within each superpixel region are averaged as the color of each superpixel region, and the number of pixels within each superpixel region is calculated to obtain a color histogram of the superpixel image.

Six superpixel algorithms are used for the superpixel segmentation of three high-resolution images obtained by the Real-ESRGAN blind super-resolution network; they are SLIC [28], linear spectral clustering (LSC) [36], WT [29], MMGR-WT [30], and the improved MMGR-WT proposed in this paper. The results are shown in Figure 4. From Figure 4b–d, it can be seen that the processing effects of SLIC, LSC, and WT exhibit over-segmentation. In Figure 4e, it can be seen that the number of superpixel regions upon using MMGR-WT is significantly reduced compared with the previous three algorithms, but there is a loss of detail information in the red boxes of the first and second subfigures. As can be seen in Figure 4e, the superpixel algorithm proposed in this paper has fewer superpixel regions due to the smoothing of the bilateral filtering process, and there is no loss of detailed information.

Figure 5a–c correspond to the number of colors in the high-resolution images, the number of colors in the superpixel images, and the color histograms of the superpixel images for the three patterns, respectively. The colors of the data points in the Figure 5a,b represent the colors of the pixel and superpixel regions, respectively. As shown in the figure, after the improved MMGR-WT pre-color-separation process, the number of colors in the images is substantially reduced, which greatly reduces the computational effort of the subsequent clustering algorithm. The histogram of the superpixel image counts the number of pixels within each superpixel region, and the image is ready for fast FCM clustering.

### 2.2. Improved DPC Algorithm

To solve the problem whereby the number-of-clusters parameter needs to be set manually, this paper uses an improved DPC algorithm to calculate the number of clusters for superpixel images without affecting the perception of the human eye. This method uses the NBS distance to calculate the color distance between superpixels and redefines the truncation distance *d_c_* according to the relationship with the perception of the human eye.

The superpixel image is still in an over-segmented state, although the number of colors has been reduced. The DPC algorithm automatically calculates the number of clusters of data points based on a decision graph, and the algorithm does not suffer from memory overflow when applied to superpixel images. However, the truncation distance parameter *d_c_* of the DPC algorithm is usually chosen empirically, and it is difficult to ensure an accurate judgment of the number of clusters. Usually, the DPC algorithm uses the Euclidean distance in calculating the distance between data points [37], and in order to measure the degree of the human eye’s perception of chromatic aberration, Miyahara et al. proposed the concept of NBS distance [38]. Instead of the Euclidean distance between data points, we determine the NBS distance by converting the superpixels from the RGB color space to the HVC space before calculating the NBS values between data points [39]. Assuming that there are two superpixels of colors A and B and their HVC values are (*H_A_ V_A_ C_A_*) and (*H_B_ V_B_ C_B_*), the formula for calculating the NBS distance can be expressed as follows:(7)$$NBS=\sqrt{2C_{A}C_{B}(1-\cos(\frac{2\pi \Delta H}{100}))+{\left(\Delta C\right)}^{2}+{\left(4\Delta V\right)}^{2}},$$
among which
(8)$$\left\{\begin{array}{l} \Delta H=\left|H_{A}-H_{B}\right|\\ \Delta V=\left|V_{A}-V_{B}\right|\\ \Delta C=\left|C_{A}-C_{B}\right| \end{array}\right.,$$

The truncation distance parameter *d_c_* is an important threshold parameter for enabling the DPC algorithm to determine whether the data points are in the same cluster class. Table 1 shows the correspondence between the NBS distance and the human eye’s perception of chromatic aberration, and when the NBS distance between two colors is greater than 3.0, the chromatic aberration of two colors is considered obvious. Therefore, we chose a truncation distance *d_c_* of 3. The local density *ρ_i_* of superpixel node *i* is calculated with a Gaussian kernel as follows:(9)$$\rho_{i}=\sum_{j=1,j\neq i}^{N}S_{j}\exp(-\frac{d_{ij}^{2}}{d_{c}}),$$
where *N* is the number of superpixels; 1 ≪ *i*, *j* ≪ *N*; *S_j_* is the number of pixels in the jth superpixel region; and *d_ij_* denotes the NBS distance between *x_i_* and *x_j_*. The larger the value of *ρ_i_*, the more likely it is to be at the center of the cluster, and on the contrary, the smaller the value of *ρ_i_*, the more likely it is to be noise or at the extremes of the dataset.

After calculating the density of each color, all the color nodes are sorted in descending order of their density values. For the color nodes, after the sorting is completed, it is also necessary to calculate the higher-density minimum distance *δ_i_* for each color node, defined as follows:(10)$$\delta_{i}=\underset{j:\rho_{j}>\rho_{i}}{\min}(d_{ij}),$$

The distance *δ_i_* is the minimum distance from *i* for all color nodes that have a higher density than color node *i*. For node *i*, the color node with higher density must be ranked in front of *i*. The NBS distances to node *i* are sequentially calculated from the 1st node to the end of the *i*-1st node, and the minimum value of the NBS distance is taken. For the color node that ranks first, the *δ_i_* value is taken as the maximum value of the NBS distance to the remaining color nodes, i.e., max(*d_ij_*). *d_ij_* is defined as:(11)$$d_{ij}=\left\|\frac{1}{S_{i}}\sum_{p\in \partial_{i}}x_{p}-\frac{1}{S_{j}}\sum_{q\in \partial_{j}}x_{q}\right\|,$$

For the superpixels whose density is the local or global maximum, their *δ* values will be much larger than those of the other superpixels, and thus, the clustering centers have larger values of ρ and *δ.* In this paper, we use the product of the local density *ρ_i_* and the distance *δ_i_* to adaptively determine the clustering center by first defining a variable *γ_i_* such that the value of *γ_i_* for a superpixel *i* is:(12)$$\gamma_{i}=\rho_{i}*\delta_{i},$$

Before calculating the *γ_i_* values of the superpixels, Equation (10) is applied to normalize *γ_i_*. The normalized data can eliminate the influence of the difference in the magnitudes of *ρ_i_* and *δ_i_* on the experimental results, and the values of *γ_i_* are mapped to the interval [0, 1] after normalization.
(13)$$\gamma_{i}=(\gamma_{i}-\gamma_{\min})/(\gamma_{\max}-\gamma_{\min}),$$

Because the *ρ* and *δ* values of the clustering centers are larger, the *γ* values of the clustering centers will also be larger. If the *γ* values of all the superpixels are ranked in order from largest to smallest, it is found that the *γ* values for superpixels that are not clustering centers are very close to each other, while the differences in the *γ* values of superpixels that are clustering centers are relatively large. Since the *γ* distribution satisfies the power law, i.e., a log function is taken for *γ*, log (γ) is approximated in a linear form. According to the optimized cutoff distance, to calculate *ρi* and *δi*, we take the log functions for *γi* and arrange them in descending order; then, we take the difference between the two adjacent numbers and find the value with the largest change in difference, and all the data points before this are recorded as the cluster center to achieve automatic calculation of the number of clusters. Using the improved DPC algorithm for the superpixel images above, the decision diagram is generated as shown in Figure 6a–c, from which the number of clusters for the superpixel images are 5, 4, and 5, respectively. 

### 2.3. Fast FCM Clustering Based on Color Histograms

To achieve fast clustering of superpixel images, we propose the following fast FCM clustering objective function based on the obtained superpixel color histogram and the number of superpixel clusters:(14)$$J_{m}=\sum_{l=1}^{q}\sum_{i=1}^{k}S_{l}u_{li}^{m}{\left\|({\frac{1}{S}}_{l}\sum_{p\in R_{l}}x_{p})-c_{i}\right\|}^{2},$$
where *l* is the color level; *q* is the number of superpixel image regions; *k* is the number of clusters calculated by the DPC algorithm; *S_l_* is the number of pixels in the lth superpixel region *R_l_*; *u_li_* is the membership intensity of the lth superpixel region belonging to the ith cluster; *m* is the weighting exponent, which is usually set to 2; *x_p_* is the pixel value of the superpixel region obtained by the improved MMGR-WT; and *c_i_* is the clustering center of the ith cluster. 

Each pixel value in the high-resolution image is replaced by the average pixel value in the corresponding region of the superpixel image, so the number of color classes is equal to the number of regions in the superpixel image. Therefore, the computational complexity of the proposed algorithm is significantly reduced since *l* ≪ *N*. *c_i_* and *u_li_* can be obtained using the Lagrange multiplier method, and the expression is as follows:(15)$$c_{i}=\frac{\sum_{l=1}^{q}u_{li}^{m}\sum_{p\in R_{l}}x_{p}}{\sum_{l=1}^{q}S_{l}u_{li}^{m}},$$
(16)$$u_{li}=\frac{{\left\|(\frac{1}{S_{l}}\sum_{p\in R_{l}}x_{p})-c_{i}\right\|}^{\frac{-2}{m-1}}}{\sum_{j=1}^{K}{\left\|(\frac{1}{S_{l}}\sum_{p\in R_{l}}x_{p})-c_{i}\right\|}^{\frac{-2}{m-1}}},$$

By updating *c_i_* and *u_li_* through Equations (15) and (16), a membership intensity matrix is obtained, and each pixel is assigned to one of the clusters with the largest membership intensity to complete the image segmentation.

The specific steps of the proposed algorithm are as follows:Set the parameters, weighting exponent *m*, iteration error *ε*, and maximum number of iterations *t*;Calculate Equations (1)–(6) to obtain the superpixel region of the image;Use Equations (7)–(13) to generate a decision graph and determine the number of clusters;Initialize the membership intensity matrix *u_li_*;Calculate the clustering center c_i_ using Equation (15);Calculate the membership intensity matrix u using Equation (16);Calculate the objective function *J_m_* using Equation (14);Determine whether $\left|J_{m}^{a}-J_{m}^{a-1}\right|\leq \varepsilon$ is valid; if so, perform step 9, and if not, return to step 4;Return the membership intensity and assign all pixels to the cluster with the largest membership intensity to complete color separation.

## 3. Experiments and Results

We chose 30 degraded patterns as experimental samples to verify the effectiveness of the algorithms in this paper. All the algorithms were run on a computer with a Win10 operating system equipped with a 2.5 GHz Intel Core i7-7300HQ CPU and 16 GB of RAM, and the experimental tool used was Matlab2016 A. 

### 3.1. Color-Separation Results of the Algorithm in This Paper

In order to demonstrate the color-separation results of this paper’s proposed algorithm, we selected three degenerate patterns from the sample. The color-separation process was carried out using the algorithm in this paper, and the results of the clarification process and the color-separation results are shown in Figure 7, Figure 8 and Figure 9.

As shown in Figure 7a, the first pattern is not particularly clear. Due to the folds in the sample fabric, there are some shadows in the image, which can affect the color-separation results. According to the perception of the human eye, the patterns have three colors, but the number of colors in the actual image is much more than three. As shown in Figure 7b, after processing by the Real-ESRGAN blind super-resolution network, a high-resolution image with clear edges is obtained, but the number of colors in the image also increases. As can be seen in Figure 7c, the color-separation algorithm proposed in this paper classifies the final image into three colors (white, red, and blue) with no color-separation errors, and the color-separation results are consistent with the perception of the human eye.

As shown in Figure 8a, the second pattern has more colors, a blurrier image, and the same folds as that in Figure 8b. It can be seen in Figure 8b that the details of the image become clear after the clarification process. In Figure 8c, the details are separated completely without any loss of information or color-separation errors. The number of colors in the final image is reduced to six, and the color-separation results are consistent with the perception of the human eye.

The third pattern has the richest colors and the most details. As can be seen in Figure 9c, the patterns are finally reduced from tens of thousands of colors to 11 colors by the color-separation algorithm proposed in this paper. The results do not show any information loss, the edges of the image are smooth and complete, and the effect is consistent with the original image.

### 3.2. Clustering Parameter Validity Test

The number of clusters is an important parameter in the color-separation algorithm proposed in this paper, and the wrong judgment of the number of clusters can lead to significant color-separation errors. In order to verify the validity of the number of clusters obtained by the improved DPC algorithm in this paper, we selected the six patterns (A~F) mentioned in the summaries of Section 2.1 and Section 3.1 as the experimental objects and evaluated the number-of-clusters parameter by adopting the clustering validity index *S_Dbw* [40].

*S_Dbw* is a density-based metric that evaluates the effectiveness of clustering by comparing the compactness within a class to the density between classes. Smaller *S_Dbw* values imply better clustering, which means that the clusters are compact and well-separated from other clusters. It is calculated using the formula:(17)S_Dbw=Scat+Dens_bw,
where *Scat* denotes intra-class tightness, and *Dens*_*bw* denotes inter-class density, defined, respectively, as follows:(18)$$S\mathrm{cat}=\frac{1}{c}\sum_{i=1}^{c}\left\|\frac{\sigma (C_{i})}{\sigma (D)}\right\|,$$
(19)$$Dens\_bw=\frac{1}{c(c-1)}\sum_{i=1}^{c}\left(\sum_{i=1i\neq j}^{c}\frac{density(u_{ij})}{\max\{density(v_{i}),dnesity(v_{j})\}}\right),$$
where *D* is the entire dataset; *c* is the number of clusters; *C_i_* represents the ith class; *σ(c_i_)* denotes the variance vector of the samples in the ith class; *σ(D)* denotes the variance vector of all the samples; *v_i_* and *v_j_* are the centers of clusters *c_i_* and *c_j_*, respectively; *u_ij_* is the mid-point of the line segment defined by the clustering centers *v_i_* and *v_j_*; and density denotes the density of the data points. 

We sequentially set the number-of-clusters parameter to 2–16 and used the color-separation algorithm proposed in this paper to color-separate the six patterns. We calculated their corresponding *S_Dbw* values, and the results are shown in Figure 10. 

A smaller value of *S_Dbw* indicates a better effect of clustering. As can be seen from Figure 10, the smallest *S_Dbw* values of patterns A–F correspond to cluster numbers of 5, 4, 5, 3, 6, and 11, respectively. Meanwhile, as can be seen above, the numbers of clusters calculated by the improved DPC algorithm in this paper are 5, 4, 5, 3, 6, and 11, which is in accordance with the numbers of colors obtained by the *S_Dbw* indicator. It can thus be proven that our algorithm can provide reliable numbers of clusters for degraded patterns and can achieve automatic color separation of patterns.

### 3.3. Comparison with Other Color-Separation Algorithms

To evaluate the superiority of the algorithms in this paper, five popular color-separation algorithms were used for comparison; they are FCM [17], DSFCM_N [19], SLIC-FCM [41], MMGR-AFCF [42], and SFFCM [30]. FCM and DSFCM_N are pixel-based clustering algorithms, and SLIC-FCM, MMGR-AFCF, and SFFCM are superpixel-based clustering algorithms. For better comparison, we added the Real-ESRGAN blind super-resolution network as a clarification step to all five algorithms. And since FCM, DSFCM_N, SLIC-FCM, and SFFCM all need manual input of the clustering number parameter, we kept this parameter consistent with the clustering number parameter calculated in this paper. Figure 9 shows the color-separation results of each algorithm for the four images used in the experiment.

According to the color-separation results, all the color-separation algorithms have some color-separation effect, but there are some detail problems, which are highlighted by red boxes. As shown in Table 2b, in the color-separation results of FCM, there are many variegated spots at the boundaries of colors. As shown in Table 2c, in the color-separation results of DSFCM_N, the first and third images have considerable color-separation errors; the colors do not match the original images, and the second image has variegated spots. As shown in Table 2d, there are no variegated spots in the color-separation results of the SLIC-FCM algorithm, but the edges are not smooth enough, and the three red areas are incorrectly concatenated into one area in the second pattern. As shown in Table 2e, MMGR-AFCF is unable to preserve the detailed parts of the image, causing serious distortion and the worst results with color-separation errors. According to Table 2f, in the color-separation results of the SFFCM algorithm, there are no variegated spots, and the edges are very smooth, but there is some detail information loss. In Table 2g, it can be noted that the color-separation algorithm proposed in this paper is able to accurately color-separate the detailed regions in the four images from the experimental results; moreover, its results have no variegated spots, and the edges of the images are smooth and accurate, without any information loss. Compared with the other five algorithms, the color-separation algorithm proposed in this paper has the best results.

In order to compare the color-separation accuracy of these six algorithms for 30 degraded patterns, we used two performance metrics, quantitative score (S) and optimal segmentation accuracy (SA) [43], and produced the corresponding ground truth (GT) [44] for clarification-processed images based on manual color separation. The quantitative score (S) is the degree of equality between the pixel set Ak and the ground truth Ck, and the optimal segmentation accuracy (SA) is the sum of correctly classified pixels divided by the total number of pixels. S and SA are defined as:(20)$$S=\sum_{k=1}^{c}\frac{A_{k}\cap C_{k}}{A_{k}\cup C_{k}},$$
(21)$$SA=\sum_{k=1}^{c}\frac{A_{k}\cap C_{k}}{\sum_{k=1}^{c}C_{j}}$$
where *A_k_* is the set of pixels belonging to the *k*th class found by the algorithm, and *C_k_* is the set of pixels belonging to the class in GT. Larger values of *S* and *SA* indicate closer results to the ground truth.

For each algorithm, for comparison, we set the values of the internal parameters to be optimal. The weighting factor for all the algorithms was 2, the iteration error was 10^−6^, and the maximum number of iterations was 100; furthermore, the number of superpixels for the SLIC-FCM algorithm was 800, and the merging radius was 1. For MMGR-AFCF, SFFCM, and the algorithm proposed in this paper, the minimum radius of the SE was *r_1_* = 1.

Thirty degraded patterns were used as experimental samples and six color-separation algorithms were used to color-separate these degraded patterns. The average scores of the corresponding segmentation performance evaluation indexes (*S* and *SA*) of each algorithm, as well as the average computation time of each algorithm, were calculated, and the results are shown in Table 3.

From Table 3, it can be seen that among the pixel-based color-separation algorithms, the FCM algorithm has higher evaluation metrics (*S* and *SA*), but the computation time is long. The DSFCM_N algorithm has the longest running time due to the computation of spatial neighborhood information in each iteration, and the segmentation accuracy of the algorithm is not high enough. Among the superpixel-based algorithms, SLIC-FCM also has a long computation time due to the high number of generated superpixels, and the color-separation effect is not satisfactory. The MMGR-AFCF algorithm has a much lower computation time than the previous three algorithms but has the lowest segmentation accuracy. The computation time of SFFCM is the best, but the accuracy of the color separation is not high. The fast, automatic fuzzy c-means color-separation algorithm based on superpixels proposed in this paper achieves an average quantitative score S of 92.11% and a segmentation accuracy of 95.78%, which are much higher than those achieved by the other algorithms, and the running time is much smaller than that of the pixel-based clustering algorithm. However, due to the addition of bilateral filtering, the running time is slightly higher than that of SFFCM. Therefore, it can be proven that the color-separation algorithm proposed in this paper is capable of accurate automatic color separation with very low computational cost for degraded patterns.

### 3.4. Comparison of Computational Complexity of Different Algorithms

The proposed color-separation algorithm can be roughly divided into three stages: the first is image clarification processing, the second is image superpixel segmentation, and the third is automatic fuzzy clustering. Since none of the other algorithms have a clarification processing step, we chose image superpixel segmentation and automatic fuzzy clustering to compare our algorithm’s computational complexity with that of the other algorithms.

The time complexity of the six algorithms is shown in Table 4. The computational complexity of SLIC is *O*(*N × K × t*′), and that of MMGR is *O*(*N × T*′), which is lower than that of SLIC. *K* denotes the number of neighboring centers, and *t*′ and *T*′ are iteration numbers and are usually less than *t*. w denotes the size of the neighboring window, and *K* denotes the number of neighboring centers. DSFCM_N has the highest computational complexity because the neighboring information is computed at each pixel. SFFCM and the algorithms in this paper have the lowest computational complexity because MMGR is fast, and the number of superpixels *N*′ is much smaller than the number of pixels *N.*

## 4. Discussion

This paper describes the significance and application value of color-separation algorithms for degraded patterns, as well as several common methods in the field of clarification and color separation. The advantages and disadvantages of super-resolution reconstruction networks, blind super-resolution reconstruction networks, pixel-based clustering algorithms, and superpixel algorithms are comprehensively and systematically analyzed. According to the application background, a complete set of color-separation algorithms for degraded patterns is proposed using the Real-ESRGAN blind super-resolution network, an improved MMGR-WT superpixel algorithm, an improved DPC algorithm, and fast FCM clustering based on color histograms, and the color-separation algorithm proposed in this paper are experimentally compared with other algorithms to prove the effectiveness of this paper’s proposed algorithm.

The experimental results show that the color-separation algorithm proposed in this paper can achieve the following objectives: (1) The clarification of degraded patterns by the Real-ESRGAN blind super-resolution network yields a high-resolution image with clear boundaries and increases the accuracy of the color-separation algorithm. (2) The improved MMGR-WT superpixel algorithm is used as a color-separation preprocessing method, which simplifies the information complexity of high-resolution images, obtains superpixel images with smooth and accurate edges, and solves the problem of variegated spots that exist in the pixel-based color-separation algorithm. (3) Using the improved DPC algorithm, the number of clusters in the superpixel image is automatically calculated without affecting the perception of the human eye, and the number of colors in the color-separation algorithm does not require manual intervention. (4) The fast clustering of superpixel images is achieved by fast FCM clustering based on a color histogram to obtain the final color-separation results.

## 5. Conclusions

With the aim of improving upon existing color-separation algorithms that deal with degraded patterns, which have unsatisfactory color-separation effects and for which the number-of-clusters parameter needs to be managed manually, a fast and automatic superpixel-based FCM color-separation algorithm is proposed. Considering that the patterns are clarified by the Real-ESRGAN blind super-resolution network, the resolution of the images is elevated, which will greatly increase the computation complexity of the color-separation algorithm. In this paper, the superpixel algorithm is used for image color separation for the first time, and the improved MMGR-WT superpixel algorithm is used as the pre-color-separation processing step, which simplifies the computational complexity of the subsequent clustering algorithm and generates superpixel images with smooth and accurate edges; the number of superpixel clusters is automatically computed by the improved DPC algorithm without affecting the perception of the human eye, which reduces human intervention; and finally, the color-separation algorithm is applied to superpixel clusters via fast FCM based on color histograms. The experimental results show that the algorithm proposed in this paper can not only achieve the automatic color-separation of degraded patterns at a very low computational cost but also attain high segmentation accuracy. 

However, we added a bilateral filtering method to the superpixel segmentation, which makes our algorithm time-consuming. In the future, we will consider using deep learning methods to train the color-separation algorithm on patterns with different resolutions to solve this problem and achieve faster and more accurate color separation.

## Figures and Tables

**Figure 1 sensors-24-00281-f001:**
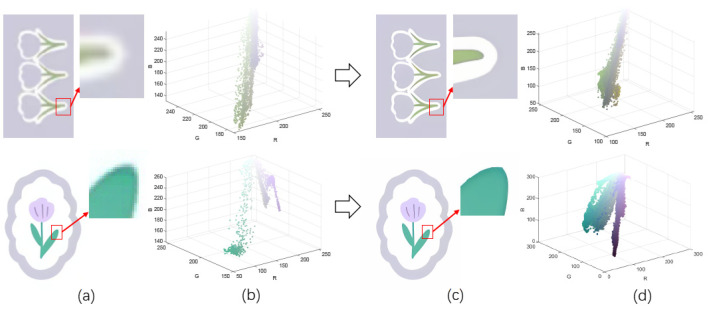
Clarification results: (**a**) Degraded patterns and their local enlargements. (**b**) Number of colors in degraded patterns. (**c**) Clarified patterns and their local magnification. (**d**) Number of colors in clarified patterns.

**Figure 2 sensors-24-00281-f002:**
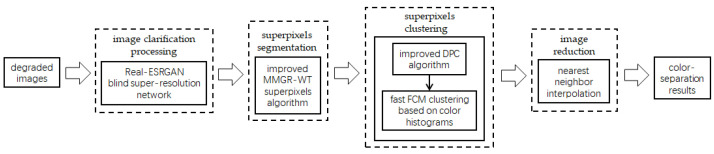
Flow of color-separation algorithm.

**Figure 3 sensors-24-00281-f003:**
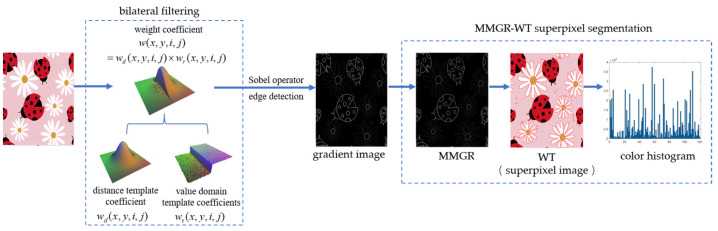
Improved MMGR-WT algorithmic framework.

**Figure 4 sensors-24-00281-f004:**
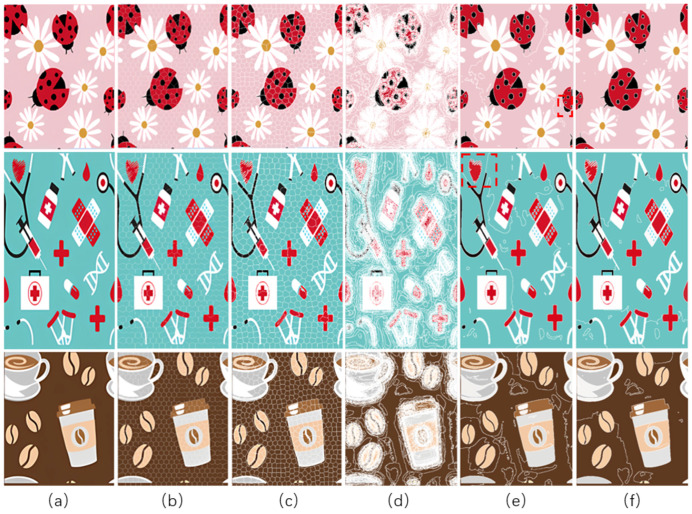
Comparison of the effects of different superpixel algorithms. (**a**) High-resolution image after clarity processing; (**b**) SLIC; (**c**) LSC; (**d**) WT; (**e**) MMGR-WT; (**f**) improved MMGR-WT.

**Figure 5 sensors-24-00281-f005:**
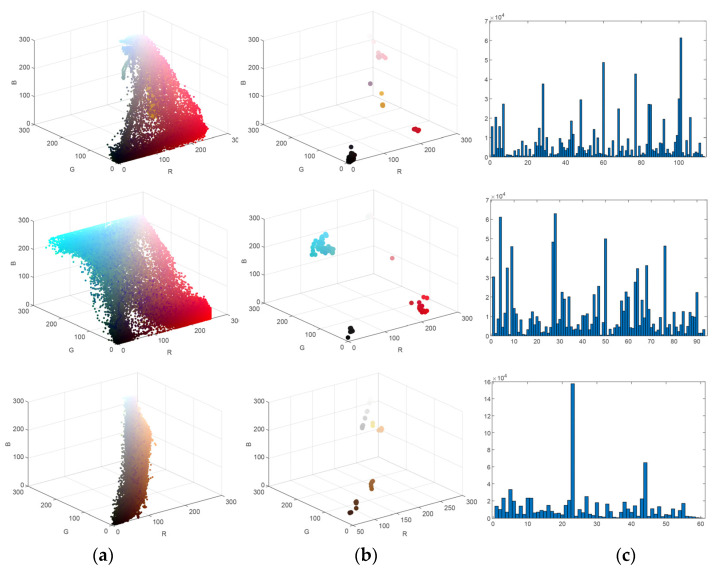
(**a**) Number of colors in the high-resolution image; (**b**) number of colors in the superpixel image; (**c**) superpixel color histogram.

**Figure 6 sensors-24-00281-f006:**
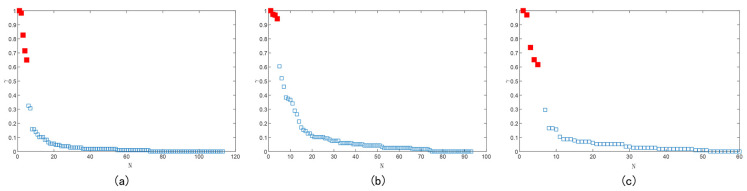
The decision diagram corresponding to the three superpixel images above.

**Figure 7 sensors-24-00281-f007:**
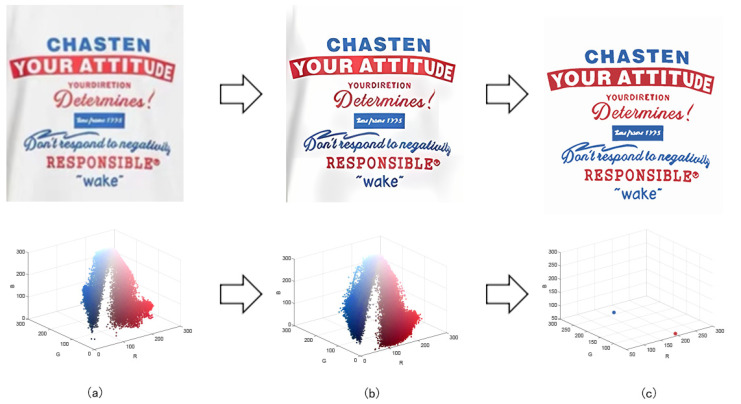
(**a**) The first pattern and its number of colors. (**b**) High-resolution image and its number of colors. (**c**) Color-separation result and its number of colors.

**Figure 8 sensors-24-00281-f008:**
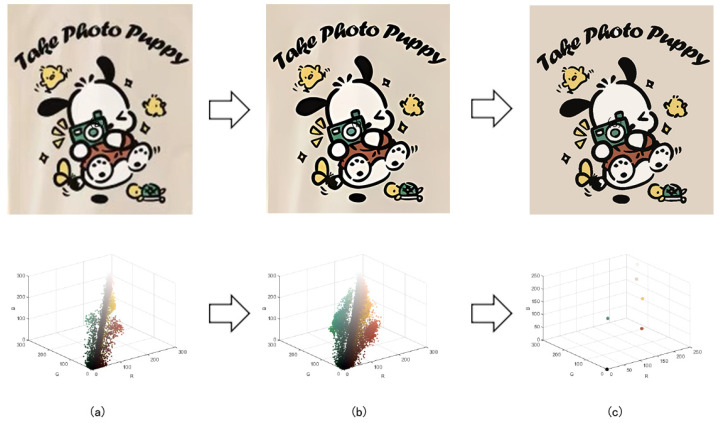
(**a**) The second pattern and its number of colors. (**b**) High-resolution image and its number of colors. (**c**) Color-separation result and its number of colors.

**Figure 9 sensors-24-00281-f009:**
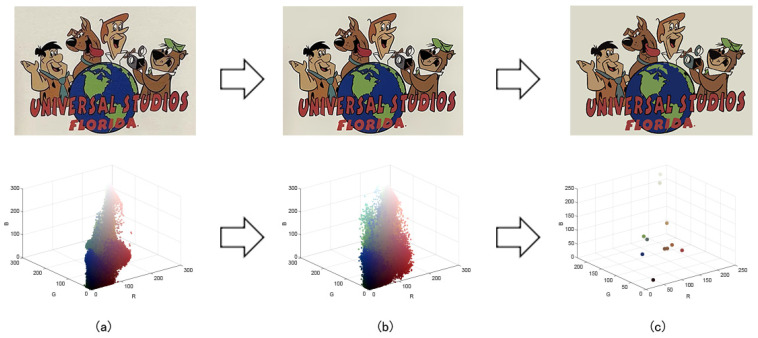
(**a**) The third pattern and its number of colors. (**b**) High-resolution image and its number of colors. (**c**) Color-separation result and its number of colors.

**Figure 10 sensors-24-00281-f010:**
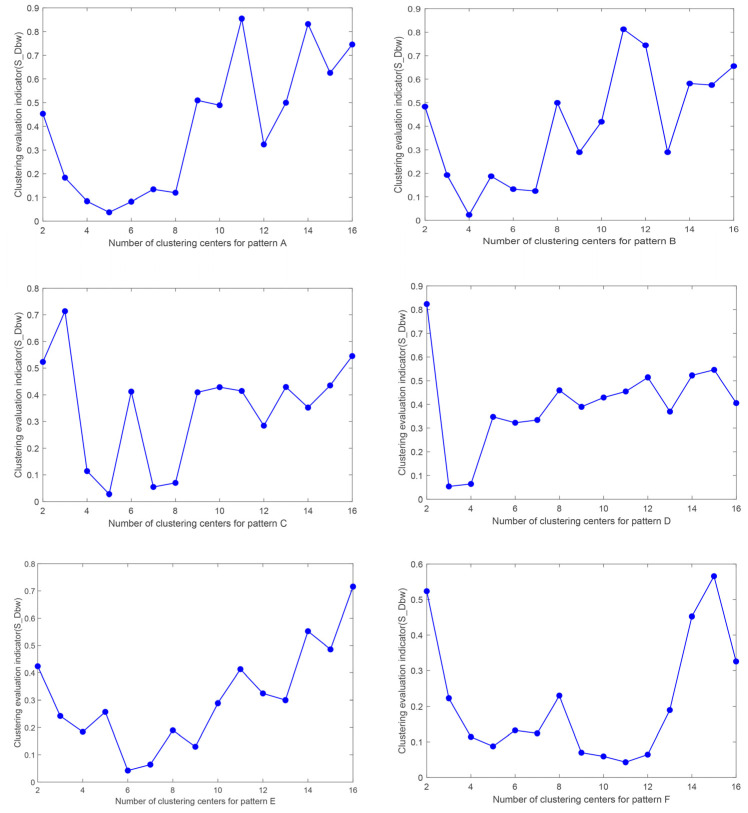
*S_Dbw* values corresponding to different number of clusters.

**Table 1 sensors-24-00281-t001:** NBS distance in relation to the perception of the human eye.

NBS Distance	0–0.5	0.5–1.5	1.5–3	3–6	6 or More
Perception of the degree of color difference	Perceived as minimal	Perceived as slight	Sense of subtlety	Perceived as obvious	Strong

**Table 2 sensors-24-00281-t002:** Comparison of various color-separation algorithms: (**a**) Degraded patterns. (**b**) Results of FCM. (**c**) Results of DSFCM_N. (**d**) Results of SLIC-FCM. (**e**). Results of MMGR-AFCF. (**f**) Results of SFFCM. (**g**) Results of this paper’s proposed algorithm.

a	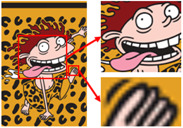	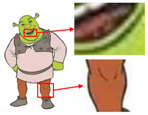	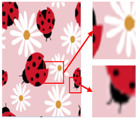	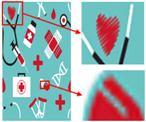
b	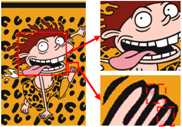	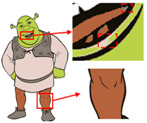	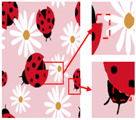	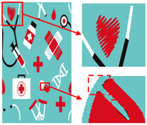
c	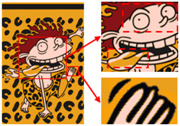	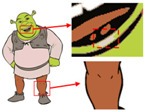	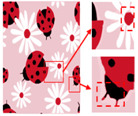	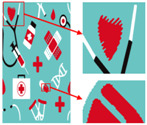
d	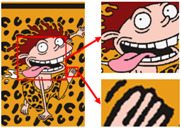	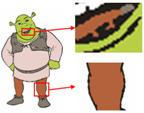	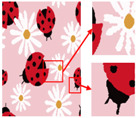	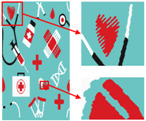
e	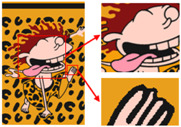	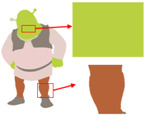	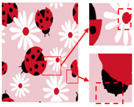	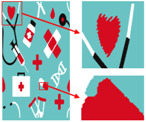
f	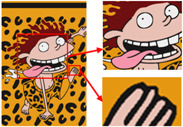	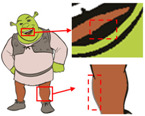	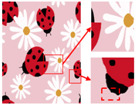	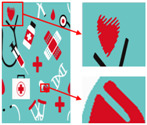
g	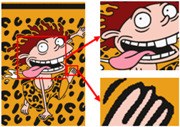	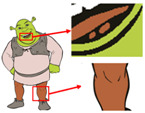	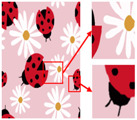	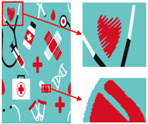

**Table 3 sensors-24-00281-t003:** Average segmentation metrics and average computation time for six algorithms for degenerate patterns.

Algorithms	Number of Samples (Amplitude)	S Mean Value (%)	SA Mean Value (%)	Average Calculation Time (s)
FCM	30	83.79	88.81	96.45
DSFCM_N	30	69.66	73.31	197.27
SLIC-FCM	30	66.45	70.31	94.32
MMGR-AFCF	30	58.62	62.33	44.39
SFFCM	30	72.32	78.88	32.14
The algorithm proposed in this paper	30	92.11	95.78	38.67

**Table 4 sensors-24-00281-t004:** Computational complexity of different algorithms.

Algorithms	Computational Complexity
FCM	O(*N × K × t*)
DSFCM_N	O(*N × w*^2^ *× K × t*)
SLIC-FCM	O(*N × K × t*′ *+ N*′ *× c × t*)
MMGR-AFCF	O(*N × T*′ *+ N*′ *× c × t × 2*)
SFFCM	O(*N × T*′ *+ N*′ *× c × t*)
The algorithm proposed in this paper	O(*N × T*′ *+ N*′ *× c × t*)

## Data Availability

Not applicable.

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
