# Peer review of "Fast Automatic Fuzzy C-Means Knitting Pattern Color-Separation Algorithm Based on Superpixels"

_sensors, 2024, doi:10.3390/s24010281_

Round 1

Reviewer 1 Report

Comments and Suggestions for Authors

In this manuscript, the authors propose a color separation algorithm based on superpixels. Overall, the presented work demonstrates certain novelty and practical value. However, the writing quality of the manuscript is relatively poor, and the following issues need to be addressed:

1. There are many obvious grammar errors in the manuscript, and it is recommended that the authors carefully proofread the entire text to correct them.

2. The authors did not provide full names or definitions when some abbreviations were initially used in the abstract or body text.

3. The linguistic quality of the manuscript needs improvement, particularly in terms of its sentence structure. The manuscript contains numerous long and complex sentences, which should be adjusted to enhance readability.

4. The citation of references is not standardized, with instances of both "[2,3]" and "[6] [7]" present simultaneously. This inconsistent citation style is not in line with the norms of academic publishing. It is recommended that the authors carefully revise and standardize the citation format throughout the manuscript.

5. To enhance the reading experience of readers, it is recommended to include hyperlinks in the manuscript for references, equations, figures, tables, sections, etc.

6. The contributions and novelties of the research need to be clearly listed at the end of the introduction.

7. Many variables are not italicized, such as lines 147 to 150.

8. The quality of some figures is poor, such as Figure 1.

9. It is recommended to adjust the layout of Figure 5 from 4x3 to 3x4 and improve the image quality.

10. There are serious problems with the representations and explanations of all equations and mathematical symbols. The authors are strongly advised to refer to high-level papers in order to understand how to properly represent and interpret equations and mathematical symbols.

11. The content of a section should be briefly introduced at the beginning of this section, such as Section 3 in line 286. This allows readers to better read and understand the relevant content of this section.

12. The figures should be centered.

13. There are a lot of low-level errors throughout the text, such as line 373.

14. Again, the authors should avoid using long sentences to improve readability. For example, "In this paper, a fast and automatic FCM color-separation algorithm based on super-pixel is proposed to address the problems that the current knitting CAD is not ideal for color-separation of degraded patterns and the number of color-separations needs to be managed manually.”

15. The conclusion obviously does not meet the requirements. The authors should present current problems, solutions, contributions, findings, limitations, and prospects for future work in a more engaging manner.

16. The authors should provide a concise discussion and description of the limitations of their current work, as well as possible future follow-up work related to it. What are the limitations of the proposed work? How should it be further improved and optimized?

Comments on the Quality of English Language

Moderate editing of English language required.

Author Response

Thank you very much for your valuable comments, here are the improvements I have made to your comments:
1、The article has been touched up and grammatical errors have been corrected
2、Abbreviations have been provided in full, as indicated by highlighting
3、Sentence structure has been adjusted
4、 "[2,3]" and "[6][7]" have been avoided, and the citation format has been adjusted.
5、 Hyperlinks have not been added for the time being
6、 Contribution and novelty have been stated in the introduction
7、The variables of the article have been corrected to italics
8、The fuzzy images in Fig. 2 (a) and (c) are the original degradation patterns of the research object, which can not be clear, and (b) and (d) have been replaced with 60 dpi images. And all other experimental images have been replaced with high quality 60 dpi images.
9、 The layout of Fig. 5 has been adjusted and divided into two parts, Fig. 5 and Fig. 6
10、Equations and mathematical expressions have all been adjusted
11、 A brief introduction has been made at the beginning of section 3, in line 330 of the text, highlighted section
12、 Figures have been centered
13、 Writing errors have been corrected, in line 445
14、 Avoided using long sentences, as in paragraph 9, highlighted section
15、 Problems, solutions, contributions, findings, limitations, and prospects for future work have been added to the conclusion, line 535, highlighted section.
16、 Limitations and follow-up work have been discussed and briefly described, line 535, highlighted section.

Reviewer 2 Report

Comments and Suggestions for Authors

Reviewer's report:

Title: Research on fast automatic FCM knitting product pattern color-separation algorithm based on superpixels

This manuscript presented a fast and automatic FCM color-separation algorithm based on superpixel. The algorithm can automatically determine the number of colors in the pattern and achieve accurate color-separation of degraded pattern. Experimental results show the proposed method’s advantages compared with four related algorithms.

Some detailed comments are as follows:

1. The last column in Table 2 listed the average computation time of five algorithms, which only refers to color-separation algorithms. In fact, the proposed method can be seen as a pipeline of several algorithms, and the former steps are all necessary for the final color-separation results. Based on this, the complexity of pipeline should be analyzed.

2. There are many parameters in the pipeline. Which are important ones? How to choose and verify the parameters estimated by the proposed method be optimal?

3. About Experiments:

(1) The proposed color-separation algorithm was compared with FCM [20], SOM-FCM [39], SLIC-FCM [40], and SFFCM [30]. These four algorithms were published in 1984, 2011, 2017 and 2018. Latest related works are advised to be compared.

(2) The manuscript illustrates the effectiveness of the proposed color-separation method by 20 degraded patterns. More data for test are encouraged.

4. From References, besides three references published in 2021, 2022 and 2023, there is no more recent works are discussed or analyzed. More latest relevant works should be considered to show the value and novelty of this study.

5. Some minor errors:

(1) Line 82: “but The superpixel image edges”, the initial letter for “The” should be lowercase;

(2) Duplicate title: “2.2. Improved DPC algorithm” and “2.3. Improved DPC algorithm”;

(3) Extra comma for Eq. (11);

(4) Line 269: “By updating ci and uli through Eq. (15) and Eq. (16)”, where are Eq. (15) and Eq. (16)?

There might be more problems in the English and phrasing in your paper. You should check the entire paper carefully and correct all of them.

Comments on the Quality of English Language

There might be more problems in the English and phrasing in your paper. You should check the entire paper carefully and correct all of them.

Author Response

Thank you very much for your valuable comments, here are the improvements I have made to your comments:
1、 I added the complexity analysis of the algorithm and compared it with other algorithms, in the position of section 3.4 of the article
2、The number of clusters parameter is the most important, added the experimental verification of this parameter, in the position of section 3.2 of the article. By using the clustering evaluation index S_Dbw, the number of clusters calculated by the algorithm of this paper is verified to be optimal, and the smaller the value of S_Dbw indicates that the clustering is more effective.
3、 two new algorithms are added as a comparison, in line 403 of [19] and [42]. However, due to time constraints, there are not many recent papers for which the code is currently available, and only these two 2019 algorithms can be found as comparisons; my experimental pattern is increased to 30 pairs.
4、References added some new papers, but it may still not be enough.
5、The article has been touched up and all the writing errors have been fixed according to your comments.

Reviewer 3 Report

Comments and Suggestions for Authors

The paper introduces a color-separation method but is overall inadequately prepared.

1. Abbreviations such as FCM, CAD, DPC, etc., are not explained, making the paper challenging to understand.

2. The significance of color-separation is not clearly presented in the Abstract or Introduction.

3. Repetition of the phrase "the improved DPC algorithm is applied to..." occurs multiple times in Lines 112, 113, 114.

4. The subtitles for sections 2.2 and 2.3 are identical.

5. In the figures, distinguishing between the high-resolution image and the color-separation result is difficult. Additional details should be included.

6. The provided examples are relatively simple; it would be beneficial to assess how the method performs on realistic images with more complex color distribution.

7. The English writing requires polishing.

I recommend a major revision for this paper.

Comments on the Quality of English Language

Please see the comments for the authors.

Author Response

Thank you very much for your valuable comments, here are the improvements I have made to your comments:
1. CAD has been explained, in lines 27-33; in lines 16, 17, and 24, in the Abstract section, the full names of FCM and DPC have been added; in the Introduction section, in lines 70-76, pixel-based color-separation algorithms, such as FCM, have been explained; and in lines 84-84, DCPC has been explained;
2. color-separation has been explained in the introduction, lines 34-45
3. the article has been touched up and writing errors fixed, e.g. lines 134-137
4. the same errors in subtitles 2.2 and 2.3 have been corrected
5. The experimental subjects have been replaced, and the results of high-resolution image and color image separation can be clearly distinguished, as in Figs. 7-9.
6、According to your suggestion, the experimental objects have been changed to patterns taken in real scenes.
7. The article has been touched up

Round 2

Reviewer 1 Report

Comments and Suggestions for Authors

With the efforts of the authors, the quality of the paper has been significantly improved. Therefore, the paper can be accepted for publication.

Comments on the Quality of English Language

Minor editing of English language required.

Author Response

Thanks

Reviewer 3 Report

Comments and Suggestions for Authors

The authors have made revisions, and the paper looks better. No other comments.

Author Response

Thanks